# Glucose Regulates m^6^A Methylation of RNA in Pancreatic Islets

**DOI:** 10.3390/cells11020291

**Published:** 2022-01-15

**Authors:** Florine Bornaque, Clément Philippe Delannoy, Emilie Courty, Nabil Rabhi, Charlène Carney, Laure Rolland, Maeva Moreno, Xavier Gromada, Cyril Bourouh, Pauline Petit, Emmanuelle Durand, François Pattou, Julie Kerr-Conte, Philippe Froguel, Amélie Bonnefond, Frédérik Oger, Jean-Sébastien Annicotte

**Affiliations:** 1Institut Pasteur de Lille, University of Lille, Inserm, CHU Lille, CNRS, U1283-UMR 8199-EGID, F-59000 Lille, France; florine.bornaque@univ-lille.fr (F.B.); clement.delannoy@univ-lille.fr (C.P.D.); emilie.courty@univ-lille.fr (E.C.); nabil14006@gmail.com (N.R.); charlene.carney@cnrs.fr (C.C.); laure.rolland@univ-lille.fr (L.R.); maeva.moreno@inserm.fr (M.M.); gromada.xavier@gmail.com (X.G.); cyril.bourouh@univ-lille.fr (C.B.); emmanuelle.durand@cnrs.fr (E.D.); p.froguel@imperial.ac.uk (P.F.); amelie.bonnefond@cnrs.fr (A.B.); frederik.oger@univ-lille.fr (F.O.); 2University of Lille, Inserm, CHU Lille, U1190-EGID, F-59000 Lille, France; pauline.petit@univ-lille.fr (P.P.); francois.pattou@univ-lille.fr (F.P.); jkerr-conte@univ-lille.fr (J.K.-C.)

**Keywords:** epitranscriptome, insulin secretion, pancreatic beta cell, type 2 diabetes

## Abstract

Type 2 diabetes is characterized by chronic hyperglycemia associated with impaired insulin action and secretion. Although the heritability of type 2 diabetes is high, the environment, including blood components, could play a major role in the development of the disease. Amongst environmental effects, epitranscriptomic modifications have been recently shown to affect gene expression and glucose homeostasis. The epitranscriptome is characterized by reversible chemical changes in RNA, with one of the most prevalent being the m^6^A methylation of RNA. Since pancreatic β cells fine tune glucose levels and play a major role in type 2 diabetes physiopathology, we hypothesized that the environment, through variations in blood glucose or blood free fatty acid concentrations, could induce changes in m^6^A methylation of RNAs in pancreatic β cells. Here we observe a significant decrease in m^6^A methylation upon high glucose concentration, both in mice and human islets, associated with altered expression levels of m^6^A demethylases. In addition, the use of siRNA and/or specific inhibitors against selected m^6^A enzymes demonstrate that these enzymes modulate the expression of genes involved in pancreatic β-cell identity and glucose-stimulated insulin secretion. Our data suggest that environmental variations, such as glucose, control m^6^A methylation in pancreatic β cells, playing a key role in the control of gene expression and pancreatic β-cell functions. Our results highlight novel causes and new mechanisms potentially involved in type 2 diabetes physiopathology and may contribute to a better understanding of the etiology of this disease.

## 1. Introduction

Glucose metabolism is physiologically regulated through a feedback loop between insulin-producing pancreatic β cells and insulin-sensitive tissues (liver, muscle, adipose tissues), in which tissue sensitivity to insulin correlates with the magnitude of the β-cell response. Type 2 diabetes (T2D) is characterized by high blood glucose levels and develops due to inadequate pancreatic β-cell function (i.e., insulin secretion) in the face of peripheral insulin resistance. Genome-wide association studies, as well as post-mortem studies of diabetic patients’ pancreas specimens, have revealed that β-cell dysfunction is thought to have a major role in the pathogenesis of T2D [1,2,3,4]. The restoration of normal β-cell mass and function has therefore become a field of intensive research seeking for the next generation of anti-diabetic drugs. Among the factors that impact the disease, obesity, physical inactivity and ageing are considered as non-genetic risk factors contributing to T2D development. These environmental factors have been reported to shape subtle and reversible modifications of DNA, named epigenetic regulations, influencing gene transcription and organ dysfunction. Although the mechanisms underlying β-cell dysfunction are still debated, emerging data suggest that epigenetic modifications are necessary for β-cell adaptation to metabolic stress [5]. Most importantly, deregulation of the epigenome (e.g., a combination of histone post-translational modifications and their interacting proteins [6]) results in the development of metabolic disorders, such as T2D [7,8,9].

The identification of specific DNA epigenetic marks and of their epigenomic modifiers that are involved in the control of β-cell function has been the purpose of tremendous effort for several decades [10]. In spite of these illuminating investigations on the epigenome, several other regulatory aspects of gene expression remain underestimated in the context of T2D. For example, RNAs are subjects to posttranscriptional modifications, through direct chemical modifications, editing or non-templated nucleotide additions [11]. The regulation of the transcriptome is the key to cellular processes that underpin cell biology, development, tissue function and diseases. It is now emerging that the modification status of the transcriptome is dynamic and responsive to environmental/developmental cues, exactly as for the DNA. This has altogether elicited the achievement of an epitranscriptome where post-transcriptional RNA modification coupled with the recruitment of effector RNA binding proteins and enzymes dynamically regulate genomic output and RNA function. Importantly, mutations in core-regulators of the epitranscriptome are key factors contributing to many human diseases or congenital disorders [12].

Epitranscriptomic modifications are dynamic and reversible modifications that affect RNAs fate. Among the 170 characterized epitranscriptomic modifications, one of the most prevalent in eucaryotes is m^6^A methylation in position N6 of adenosine [13]. The methyl group is added in a co-transcriptional manner [14,15], with the use of S-adenosylmethionine by a complex of proteins composed of the two methyltransferases METTL3 and METTL14 [16], associated with the proteins WTAP [17], KIAA1429 [18], RBM15/15B [19], ZC3H13 [20], and VIRMA [21] that facilitate the localization of the complex on specific regions of the RNA. This process is reversible since m^6^A methylation can be removed by two eraser enzymes, the Fat mass and obesity-associated protein (FTO) and ALKBH5 demethylases. FTO, being first identified as a susceptibility gene for obesity [22] and T2D, can remove m^6^A methylation from RNA in the nuclear speckles [23], whereas ALKBH5 specifically demethylates m^6^A-methylated RNAs in the nucleus [24]. This methylation is mainly found in the CDS and 3′UTR regions and around the stop codon, in a consensus sequence named DRACH (D = G/A/U; R = G/A; H = AUC) [25,26,27]. Once modified by m^6^A methylation, RNAs are recognized by reader proteins, which will affect the translation [28], splicing [29], stability [30], structure [31] or export of RNAs from the nucleus to the cytoplasm [32]. m^6^A methylation of RNAs plays critical roles in many physiological processes [19,33,34] but also contributes to disease development such as T2D [35]. Recent studies have demonstrated that the genetic invalidation of *Mettl3* and/or *Mettl14* in mouse pancreatic β cells leads to loss of β-cell identity, function and mass, associated with decreased insulin secretion and hyperglycemia [35,36,37,38]. Other studies have described a decrease in m^6^A methylation in the pancreatic islets of patients with type 2 diabetes, associated with a decreased expression of the methyltransferases *METTL3* and *METTL14* [35]. Finally, decreased levels of m^6^A methylation in blood sample of patients with type 2 diabetic are associated with increased expression levels of the demethylase *FTO* [39,40] that correlates positively with blood glucose concentration [40]. Although these studies identified a link between the epitranscriptome and T2D, the underlying molecular mechanisms involved in the modulation of m^6^A levels within pancreatic β cells in physiological and pathological conditions remain unknown.

Here, we tested the hypothesis that the metabolic environment of the pancreatic β cell could directly modulate the epitranscriptome through the regulation of m^6^A methylation of RNAs. We show in this study that m^6^A levels are decreased in T2D human islets and are regulated by glucose. We also demonstrate that the cellular localization of epitranscriptomic enzymes is regulated by glucose, suggesting that their activity is dependent on glucose concentrations. Finally, we demonstrate that blocking these enzymes has a strong impact on β-cell identity and functions, highlighting a molecular link between glucose, m^6^A and pancreatic β-cell function.

## 2. Materials and Methods

### 2.1. Animal Experiments

Mice were housed according to European Union guidelines for the use of laboratory animals. In vivo experiments were performed in compliance with the French ethical guidelines for studies on experimental animals (animal house agreement no. 59-350294, authorization for animal experimentation, project approval by our local ethical committee no. APAFIS#2915-201511300923025v4). C57BL/6J mice were maintained under 12 h light/dark cycle. 5-weeks-old male mice were fed with normal diet or high fat diet (60% of fat, Research Diet) for 12 weeks. Metabolic phenotyping, including measurements of body weight, glycemia and glucose tolerance, was performed as described previously [8]. db/db mice (strain BKS.Cg-*Dock7^m^+/+Lepr^db^J*) were purchased from Charles River Laboratories (Wilmington, MA, USA).

For mouse pancreatic islet isolation, pancreas was injected with 1.5 mg/mL of type V collagenase (C9263, Sigma, St. Louis, MO, USA) and digested at 37 °C during 10 min. After filtration with a sieve, islets were purified with polysucrose density gradient medium (from histopaque 1119, Sigma) of 1119, 1100, 1080 and 1060 g/mL. Finally, pancreatic islets were sampled between the 1100 and 1080 gradient and then handpicked under binocular microscope for better purity. Mouse pancreatic islets were then incubated during 24 h at 37 °C and 5% CO_2_, in culture medium composed of RPMI 1640 and Glutamax (61870044, Gibco, Waltham, MA, USA) supplemented with 10% fetal bovine serum and 100 pg/mL penicillin-streptomycin. For Fto inhibitor treatment, mouse pancreatic islets were treated during 2 h under the same conditions as described for Min6 cells (see Section 2.3).

### 2.2. Human Pancreatic Islets Studies

Human pancreatic islets were provided by Pr. F. Pattou in the context of ECIT, JDRF funded islet distribution program and isolated according to Kerr-Conte et al., 2010 [41]. Islets were harvested from brain-dead, non-diabetic and type 2 diabetic adult human donors (Appendix A for donor information). For palmitate treatment, islets were incubated with 5.6 mM or 20 mM glucose with or without 0.5 mM sodium palmitate coupled with FFA-free human serum albumin during 72 h, with daily change of the medium. For treatment with the FTO inhibitor, pancreatic islets were incubated at the indicated time with 100 nM of bisantrene (B4563, Sigma). Pancreatic islets were then snap-frozen for further processing. For glucose-stimulated insulin secretion (GSIS) tests, 1 h before the end of the bisantrene treatment, human islets were starved in Krebs–Ringer bicarbonate buffer supplemented with 0.5% BSA and 100 nM of bisantrene and processed for GSIS as described below.

### 2.3. Cell Culture and Treatments

Min6 cells (Addexbio, San Diego, CA, USA) were cultured in complete Min6 medium containing DMEM medium with 4.5 g/L of glucose (31966-021, Gibco) supplemented with 15% fetal bovine serum, 100 µg/mL penicillin-streptomycin and 55 µM β-mercaptoethanol. Cells were maintained at 37 °C and 5% of CO_2_. For acute glucose treatment, Min6 cells were cultured in complete Min6 medium with 2.8 or 20 mM of glucose for 1, 2, 3 or 6 h. For long-term treatments, Min6 cells were incubated with 5.6 or 20 mM glucose with or without 1 mM palmitate coupled with FFA-free bovine serum albumin (BSA, A7030, Sigma) during 72 h, with a daily change of the medium. For treatments with the FTO inhibitor, cells were incubated with 100 nM of bisantrene (B4563, Sigma) in complete Min6 medium. 1 h before the end of treatment, Min6 cells were starved in Krebs–Ringer bicarbonate buffer supplemented with 0.5% BSA and 100 nM of bisantrene for glucose-stimulated insulin secretion tests.

### 2.4. siRNA Transfection

The antibiotics were first removed from the medium by 2 washes with DMEM. Cells were then transfected with Dharmafect 1 (Horizon Discovery, Waterbeach, UK) and 25 nM of siRNA (Smartpool ON-TARGETplus siRNA, Horizon Discovery) in complete Min6 medium without antibiotics following manufacturer’s instructions. Cells were lysed 48 h after treatment and further processed for RNA or protein extraction (See Appendix A for siRNA information).

### 2.5. Glucose-Stimulated Insulin Secretion and Insulin Quantification

For GSIS, Min6 cells, mouse and human pancreatic islets were starved for 1 h with Krebs-Ringer bicarbonate buffer supplemented with 0.5% Bovine Serum Albumin (BSA). Cells were then stimulated with 2.8 or 20 mM of glucose for 1 h. Insulin secretion and insulin content (obtained after lysis with 180 mM HCl and 75% ethanol, then neutralized by 1M Na_2_CO_3_) were assessed through ELISA (mouse or human Elisa insulin kit, Mercodia, Winston Salem, NC, USA). Insulin secretion was then normalized to insulin content and expressed as a percentage of insulin content or as a stimulation index corresponding to the fold of insulin secretion at 20 mM over 2.8 mM of glucose.

### 2.6. RNA Extraction and RT-qPCR

Total RNA was extracted from Min6 cells using Nucleospin RNA Kit (Macherey Nagel, Allentown, PA, USA) according to manufacturer’s instructions. RNA from isolated pancreatic islets were extracted using Rneasy Micro Kit (Qiagen, Hilden, Germany). RNA was retrotranscribed using SuperScript III reagents (Invitrogen, Waltham, MA, USA), random hexamers and dNTPs. Expression of mRNA was quantified by real-time qPCR using gene-specific oligonucleotides (See Appendix A for oligonucleotide information) and FastStart SYBR Green Master Mix (Roche, Basel, Switzerland). Results from Min6 cells and mouse pancreatic islets were normalized to mouse cyclophilin mRNA levels. Data from human islets were normalized to human cyclophilin mRNA levels.

### 2.7. m^6^A Quantification by Dot Blot

500 ng of RNAs was used for m^6^A quantification through dot blot experiments. Briefly, RNAs were blotted on a nylon membrane using a dot blotter with aspiration for 15 min and fixed by UV with 130 J/cm² (254 nm). Membranes were incubated with 0.04% methylene blue and 0.5 M sodium acetate to mark total RNA. Before antibody incubation, membranes were washed with TBS-Tween 0.1% and blocked 1 h at room temperature with 5% nonfat milk in 0.1% TBS-Tween. A primary anti-m^6^A antibody was used to mark m^6^A methylation of RNAs for an overnight incubation at 4 °C (See Appendix A for the antibodies used in this study). After washing the membrane with 0.1% TBS-Tween, they were incubated with secondary antibodies conjugated with horseradish peroxidase. m^6^A-methylated RNAs were then visualized with enhanced ChemiLuminescence Western blotting substrate (Pierce, Waltham, MA, USA). Labeling of m^6^A-methylated RNA was normalized to total RNA using the methylene blue labeled membrane.

### 2.8. m^6^A Quantification by ELISA

m^6^A quantification was done using the EpiQuik m^6^A RNA methylation Kit (Colorimetric, Epigentek, Farmingdale, NY, USA) according to the manufacturer’s instructions, using 200 ng of total RNA and primary antibody against m^6^A modification.

### 2.9. Immunofluorescence and Quantification

Immunofluorescence analysis were performed on pancreatic tissues that were fixed in 10% formalin, embedded in paraffin and sectioned at 5 µm as described previously [8]. Briefly, for immunofluorescence microscopy analyses, after antigen retrieval using citrate buffer, 5-µm formalin-fixed paraffin embedded (FFPE) pancreatic sections were incubated with the indicated antibodies (See Appendix A for the antibodies used in this study). Immunofluorescence staining was revealed by using a fluorescein-isothiocyanate-conjugated anti-rabbit (for m^6^A), alexa-conjugated anti-mouse (for glucagon) or anti-guinea pig (for insulin co-staining with m^6^A) secondary antibodies. Nuclei were stained with DAPI.

After different treatments, Min6 cells were fixed with 4% paraformaldehyde in PBS during 15 min at room temperature and washed 3 times with PBS. Cells were then permeabilized with 1% PBS-Triton in the same condition. Cells were blocked in 1% PBS-BSA for 30 min at room temperature and incubated with primary antibodies for 16 h at 4 °C (See Appendix A for the antibodies). Secondary antibodies conjugated with fluorophores (AlexaFluor secondary antibodies, ThermoFisher, Waltham, MA, USA) were used to reveal immunofluorescence staining of target proteins. DAPI was used to stain nuclei. Images were processed and quantified using Macro on ImageJ Software.

### 2.10. Nuclear/Cytoplasmic Fractionation, Western Blot Analysis and Quantification

Western blot analysis was performed as previously described [42]. Briefly, after different treatments as indicated, Min6 cells were washed twice with cold PBS supplemented with phosphatase inhibitors. Nuclear/cytoplasmic fractionation was performed using Nuclear Extract Kit (40010, Active Motif, Carlsbad, CA, USA).

Immunoblotting experiments were performed using 15 μg of total proteins. After electromigration, proteins were transferred on nitrocellulose membrane during 1 h at 110 V that were further incubated in TBS-Tween 0.05% supplemented with 5% of milk. Membranes were incubated 16 h at 4 °C with primary antibodies as indicated in blocking buffer supplemented with 3% of bovine serum albumin or milk. After washing, membranes were incubated 1 h with the secondary antibody conjugated with horseradish peroxidase. The revelation of luminescent bands was performed using Pierce ECL Western blotting substrate or SuperSignal West Dura Extended duration substrate (ThermoFisher) with Chemidoc Xrs+. Quantification of band signals was performed using ImageJ software.

### 2.11. Statistical Analysis

All data are expressed as mean ± SEM. Statistical analysis were performed using GraphPad Prism 9.0 software with Mann–Whitney tests, Kolmogorov–Smirnov tests, Multiple t-tests, 1-way ANOVA with Dunnett’s post-hoc tests or 2-way ANOVA with Bonferroni’s or Tukey’s correction for multiple comparisons as indicated in the figure legends. Differences were considered statistically significant at *p* value < 0.05 (* *p* < 0.05, ** *p* < 0.01, *** *p* < 0.001, **** *p* < 0.0001).

## 3. Results

### 3.1. m^6^A RNA Methylation Is Reduced in Human and Murine Pancreatic Islets

To evaluate the potential changes in the m^6^A methylation levels in pancreatic islets during T2D, several pathophysiological models were used, in which methylation of m^6^A was quantified from isolated pancreatic islets total RNA or immunofluorescence assays using formalin-fixed paraffin-embedded (FFPE) mouse pancreata. First, m^6^A methylation of RNAs has been quantified in pancreatic islets isolated from C57Bl6/J mice fed with a control chow (CD) or with a high-fat diet (HFD) during 12 weeks. HFD feeding induced weight gain (Appendix A), increased fasting hyperglycemia (Appendix A) and impaired glucose tolerance (Appendix A) when compared to age-matched chow-fed mice. In this hyperglycemic context, we observed a 50% decrease (*p* = 0.0635) in the m^6^A methylation in pancreatic islets of mice fed with HFD compared to CD (Figure 1A). Most importantly, these results were further confirmed in human pancreatic islets obtained from donors with T2D compared to non-diabetic controls (Figure 1B, mean age ± SD, 41.2 ± 18.2 for controls versus 61 ± 6.0 for T2D; *p* = 0.07). Indeed, m^6^A methylation is reduced by more than 60% (*p* = 0.008) in islets from donors with T2D compared to non-diabetic islets. Finally, we compared m^6^A levels in FFPE pancreata obtained from 15-week-old mice fed a CD, a HFD or from the diabetic and obese mouse model db/db. Although it is well accepted that HFD represents a model of β-cell compensation during insulin resistance more than β-cell failure per se, it has been previously reported that the loss of β-cell mass and function is observed in 15-week-old db/db mice when compared to younger db/db mice [43]. Here again, we observed a decrease of immunofluorescent staining for m^6^A methylation in HFD-fed, an effect that was exacerbated in pancreas from db/db mice when compared to age-matched, CD fed wild type controls (Figure 1C). These results suggest that m^6^A levels are sensible to a hyperglycemic, T2D environment and suggest a possible role of m^6^A methylation in the loss of β cell functions during T2D.

### 3.2. High Glucose Concentrations Reduce m^6^A Methylation in Mouse β-Cell Line and Human Pancreatic Islets

Since T2D is characterized by a chronic hyperglycemia, we then investigated the possible role of modulating glucose concentrations to better understand the potential mechanisms involved in the dynamic regulation of m^6^A methylation observed in pancreatic islets from donors with T2D. Dot blot experiments demonstrate that a treatment with 20 mM glucose induced a 20% to 40% decrease in m^6^A methylation after 2 and 3 h of treatment compared to 2.8 mM glucose treatment in the mouse pancreatic β cell line Min6 (Figure 2A,B). Immunofluorescence analysis of m^6^A methylation of RNA in Min6 cells further confirmed that 20 mM glucose incubation led to a global decrease of m^6^A methylation both in the cytoplasm and in the nucleus, when compared to 2.8 mM glucose concentration (Figure 2C,D). Interestingly, in human pancreatic islets isolated from control, non-diabetic donors, we observe that treating those islets for 1 h with 20 mM glucose also induced a decrease of 20 to 30 % in m^6^A RNA methylation compared to a 2.8 mM glucose treatment (Figure 2E,F). Altogether, these data suggest that high glucose concentrations similarly decrease m^6^A methylation of RNAs in human pancreatic islets and mouse β-cell lines.

### 3.3. Glucose Induces Changes in mRNA Expression and Protein Localization of Mettl3 and Alkbh5 in Mouse Min6 Cells

The decrease in global m^6^A methylation observed after a glucose treatment could be mediated by the loss of expression or activity of m^6^A methyltransferases and/or an increased expression or activity of m^6^A demethylases. Thus, we studied the expression of selected methyltransferases and demethylases involved in the regulation of m^6^A methylations in the same conditions than described above. Upon 20 mM glucose, we observed that the mRNA levels of *Alkbh5* and *Fto* encoding demethylases and of *Mettl3* encoding methyltransferase were increased by 45%, 35% and 40%, respectively (Figure 3A–C), suggesting that the decrease in m^6^A methylation observed after glucose treatment may possibly be regulated by a rise in the mRNA expression of demethylases, which could be compensated by a concomitant increased expression of Mettl3. The increased mRNA levels of m^6^A enzymes were not associated to a rise of the expression of the early glucose responsive genes *Egr1*, *Ier2*, *Id2* and *Tob1* (Appendix A), suggesting that these genes were not directly involved in the transcriptional regulation of *Alkbh5*, *Fto* and *Mettl3* by glucose. In addition, the increase in mRNA levels was not associated to a concomitant increase in protein levels since short-term, acute treatment of Min6 cells with 20 mM of glucose for 3 or 6 h did not induce a rise in METTL3 and ALKBH5 protein levels compared to 2.8 mM of glucose concentration (Appendix A). This suggested that the enzymatic activity or nuclear localization of these enzymes, rather than their protein levels, could mediate the decrease in global m^6^A methylation upon acute glucose treatment. Therefore, since the enzymatic activity of these enzymes could also depend on their subcellular localization, we analyzed the effect of glucose on their nuclear and cytoplasmic shuttling. Although we could not detect significant changes in nuclear versus cytoplasmic localization of these enzymes through classical biochemical fractionation approaches (Appendix A), immunofluorescence analysis demonstrated that glucose treatment induced variations in the localization of ALKBH5 and METTL3. Indeed, the ALKBH5 protein, which is predominantly cytoplasmic, increased its nuclear localization upon 20 mM glucose treatment (Figure 3D,E). Conversely, the METTL3 protein, which is mainly nuclear in basal conditions, acquires also a cytoplasmic localization upon glucose treatment (Figure 3F,G). Altogether, these data suggest that glucose not only increases the mRNA levels of *Mettl3*, *Fto* and *Alkbh5* but also induces variations in the localization of those m^6^A enzymes, probably allowing access of demethylases to methylated mRNAs in the nucleus.

### 3.4. A Combination of Glucose and Palmitate Treatment Increases m^6^A Methylation in Pancreatic β Cell Line and Human Islets Associated to Alkbh5 Downregulation

In addition to chronic hyperglycemia during T2D, insulin resistance also increases the level of circulating free fatty acids (FFA), leading to dyslipidemia and an increased risk of cardiovascular disease [44]. This rise in circulating FFA contributes to β-cell dysfunctions and insulin secretion defects [45,46]. We therefore tested whether free fatty acids could also modulate m^6^A RNA methylation by treating our different experimental models with palmitate, a saturated fatty acid previously shown to trigger β-cell stress and insulin secretion defects [47]. We first tested the efficacy of a combination treatment of 1 mM palmitate and 20 mM glucose for 72 h on impairing insulin secretion and inducing the expression of genes involved in ER stress, previously known to be regulated by a lipotoxic stress [45]. As expected, the palmitate treatment reduced glucose-stimulated insulin secretion in Min6 cells (Figure 4A). In the absence of palmitate, we observed that insulin secretion represented approximately 36% of the intracellular insulin content whereas it decreased to approximately 23% after a 72 h treatment with 1 mM of palmitate. It has been previously shown that palmitate induces the expression of ER stress genes [48]. qPCR experiments demonstrated a 50% increase of the mRNA levels of *Atf4* and the spliced form of *Xbp1*, without changing the levels of *Chop* (Figure 4B). Through immunofluorescence analysis, we then studied the effect of palmitate on m^6^A methylation and on the expression of m^6^A regulating enzymes in Min6 cells (Figure 4C,D). We observed that palmitate induces an increase of 70% of m^6^A methylation of total RNA in Min6 cells. Our results also show a significant effect on decreasing the expression of *Alkbh5* demethylases after palmitate treatment (Figure 4E), which could partly explain the increase in m^6^A methylation. Although a short-term treatment with glucose had no effect on ALKBH5 and METTL3 protein levels (Appendix A), chronic treatment with glucose induced an increase in the expression of both proteins, an effect that was abolished by palmitate stimulation (Figure 4F,G). We then carried out the same experiment in human pancreatic islets (Figure 4H). The analysis of the expression of m^6^A enzymes by qPCR showed a 20% decreased expression of *ALKBH5* in the presence of palmitate. Together, these results suggest that a chronic treatment combining glucose and palmitate increases m^6^A RNA methylation in Min6 cells and human pancreatic islets, probably through reducing the mRNA and/or protein levels of the ALKBH5 demethylase.

### 3.5. Chronic Palmitate Treatment Does Not Modulate m^6^A Methylation in Pancreatic β Cell Line and Human Islets at Low Glucose Concentration

To better appreciate the metabolic factor triggering the modulation of m^6^A RNA methylation upon cotreatment with palmitate and glucose, we treated Min6 cells and human islets for 72 h with low glucose (5.6 mM) with or without palmitate. Similar to Min6 cells treated for 72 h with 20 mM glucose (Figure 4A), incubating Min6 cells with 5.6 mM glucose and 1 mM palmitate for 72 h also impaired GSIS (Figure 5A). Interestingly, palmitate, upon low glucose concentration, did not induced ER stress, as demonstrated by the analysis of *Xbp1s*, *Atf4* and *Chop* mRNA expression levels through qPCR experiments (Figure 5B). In this metabolic context, m^6^A methylation of total RNA in Min6 cells, as well as the expression levels of enzymes regulating this epitranscriptomic mark were not changed by palmitate (Figure 5D,E). These results were confirmed in human islets, in which palmitate decreased GSIS without altering *ALKBH5*, *FTO* or *METTL3* levels (Figure 5F,G). These data demonstrate that the chronic cotreatment with palmitate at a low glucose concentration may not be sufficient to modulate m^6^A methylation and/or m^6^A enzymes. Altogether, our results obtained upon low (Figure 5) and high (Figure 4) glucose concentration suggest that a chronic treatment combining 20 mM glucose and palmitate has a major effect on regulating m^6^A levels and gene expression of m^6^A related-enzymes.

### 3.6. m^6^A Enzymes Regulates the Expression of Genes Involved in the Function and Identity of Pancreatic β Cells

In order to study the potential role of those enzymes involved in m^6^A methylation of RNA within the pancreatic β cell, we transfected Min6 cells with different siRNAs targeting these enzymes. After 48 h of treatment, we observe that these siRNAs were effective against their respective target, with at least a 50% reduction of their respective transcripts (Figure 6A,D,G). In addition, we observed that the siRNA directed against *Mettl3* induced a concomitant reduction in the expression of the *Fto* demethylase of about 40%, which suggests a compensatory process (Figure 6G). Subsequently, we studied the expression of genes involved in maintaining the identity and function of the pancreatic β cell, including genes involved in the transcriptional regulation (*Foxo1*, *MafA*, *Pdx1*, *Pax4* and *Pcaf/Kat2b*) or the insulin secretion process (*Gck*, *Glut2/Slc2a2*, *Kir6.2/Kcnj11*, *Ins1* and *Ins2*).

For siRNAs directed against *Alkbh5*, we observed a slight decrease in the expression of the key marker of β cell identity, *MafA* (Figure 6B). In addition, an increase of around 30% in the expression of glucokinase (*Gck*) and *Kir6.2* was found upon *Alkbh5* knock-down (Figure 6C). Knocking down *Fto* through siRNA resulted in a significant increase of *Pax4* and *Pcaf* mRNA expression levels, as well as an increase in *Gck* and *Glut2* levels (Figure 6F). Conversely, *Ins1* mRNA levels were decreased in *Fto*-deficient Min6 cells (Figure 6F). Finally, transfecting Min6 cells with siRNAs targeting *Mettl3* mostly induced a halving of *Foxo1*, *MafA* and *Glut2* mRNA expression levels (Figure 6H,I). Altogether, our data suggest that siRNAs directed against demethylases rather induce an increase in the expression of the genes involved in the identity and function of pancreatic β cells, while those directed against the *Mettl3* methyltransferase rather decreased the expression of identity genes and may have contributed to maintaining the identity and function of pancreatic β cells.

### 3.7. Knock-Down or Pharmacological Inhibition of FTO Stimulates Glucose-Induced Insulin Secretion

To assess the potential effects of the FTO demethylase on pancreatic β cell function, we knocked down the *Fto* gene in Min6 cells using gene-specific siRNA and evaluated glucose-stimulated insulin secretion. 48 h after a treatment with the *Fto* siRNAs, we observed that knocking-down *Fto* expression in Min6 cells increased insulin secretion in response to 20 mM glucose (Figure 7A). Since the genetic depletion of *Fto* in Min6 cells increased β-cell identity and function genes (Figure 6), as well as GSIS (Figure 7A), we put forth the hypothesis that the pharmacological inhibition of this enzyme could also modulate β-cell functions. We took advantage of the recently described FTO inhibitor bisantrene to block FTO enzymatic activity [49]. We observed that treating Min6 cells with bisantrene potentiated insulin secretion in response to 20 mM glucose (Figure 7B).

This effect was further confirmed in isolated mouse islets. Indeed, treating pancreatic islets isolated from lean, non-diabetic C57Bl/6J mice with bisantrene potentiated insulin secretion after a 20 mM glucose stimulation (Figure 7C, *p* = 0.0112). We then treated human pancreatic islets to evaluate the potential of bisantrene to stimulate insulin secretion in response to glucose. Although 1-h treatment with 100 nM bisantrene slightly, albeit not significantly, increased m^6^A methylation of total RNA from human islets, no effects of bisantrene on global m^6^A methylation were observed after 4 h or 24 h treatments (Figure 7D,F). Interestingly, treating human islets with bisantrene for 24 h significantly increased glucose-stimulated insulin secretion (Figure 7G). Altogether, our data show that the genetic depletion and pharmacological inhibition of FTO potentiate glucose-stimulated insulin secretion, both in mouse and human pancreatic islets, suggesting that FTO could play a critical role in the negative regulation of insulin secretion.

## 4. Discussion

Here, we report that m^6^A methylation is reduced in pancreatic islets isolated from mice subjected to a metabolic stress and from the obese and diabetic leptin receptor-deficient db/db mouse model. More interesting is our observation that m^6^A methylation of RNAs is strongly decreased in human pancreatic islets from T2D patients, in line with recent reports showing a decrease in m^6^A methylation in human samples [35,39,40]. Altogether our data highlight that the diabetic status may contribute to m^6^A demethylation of RNAs within pancreatic islets, and probably within the insulin-producing β cells. To better understand the potential molecular mechanisms linking hyperglycemia, hyperlipidemia and m^6^A RNA methylation, we developed in vitro strategies to model hyperglycemia and study the regulation of m^6^A methylation. Here we show that exposure of pancreatic islets or mouse β-cell lines to high glucose concentration strongly decreases m^6^A RNA methylation in our pancreatic β-cell models. Strikingly, recent data obtained on diabetic human blood samples have shown a negative correlation between glucose concentration and m^6^A methylation of RNAs [40]. We can speculate that the effect of a diabetic environment, characterized by chronic hyperglycemia, on m^6^A methylation of RNAs, could therefore be mediated, at least in part, by glucose.

During T2D, the expression of *FTO* is increased and positively correlates with glucose concentration [39,40]. The *FTO* locus is a predisposing gene for obesity [22]. In mice, its genetic deficiency induces, in white adipose tissue, an increase in lipid catabolism and carbohydrate metabolism as well as a reduction of the fat mass [50]. Conversely, *Fto* overexpression causes an increase in food intake and fat mass [51]. In humans, a variant of the gene inducing its overexpression causes an increase in food intake, due to reduction in RNA methylation of ghrelin, an orexigenic hormone, and causes a decrease in its translation [52]. FTO demethylase therefore plays an important role in the regulation of metabolism, in particular in adipose tissue. It would therefore be interesting to develop a mouse model invalidated for *Fto* specifically in pancreatic β cells in order to study its role in the regulation of pancreatic β-cell metabolism. Interestingly, besides *Fto*, we also observed an increased expression of the *Alkbh5* and *Mettl3* enzymes upon high glucose concentrations. In addition, we identified that glucose modifies the subcellular localization of these enzymes, with an increase in the ALKBH5 protein in the nucleus, the main site of m^6^A methylation of RNAs as well as an increase in METTL3 in the cytoplasm, where methylation activity is less important [53]. Regretfully, due to the lack of specific and reliable anti-FTO antibodies, we could not evaluate its cellular localization upon glucose treatment. Although the precise function of ALKBH5, FTO and METTL3 in pancreatic β cells remains unknown, as well as their specific function and their target RNAs in the cytoplasm or nucleus, it is tempting to speculate that these enzymes may play a fundamental role in pancreatic β-cell function or identity, and, consequently, T2D physiopathology. Indeed, the use of siRNA has allowed us to better appreciate their roles in the regulation of pancreatic β-cell genes. In addition, we observed different effects on protein levels of these enzymes upon acute versus chronic glucose treatment. Although short-term glucose treatment induced their mRNA levels, despite changing the expression level of early glucose induced genes such as *Egr1*, *Ier2*, *Id2* and *Tob1*, changes in nuclear/cytoplasmic localization and decreased m^6^A methylation, a chronic glucose treatment rather increased their global protein levels. Whether the regulation by glucose of m^6^A enzymes is direct or indirect, involves other glucose-responsive transcriptional regulators or modulates m^6^A-dependent or independent mRNA half-life remains to be investigated. Our data provide new insights as to the potential role of the m^6^A machinery in the adaptive response of the β-cell to acute *versus* chronic stress conditions.

We found that siRNAs directed against demethylases induced an increase in the expression of genes mainly involved in the function of pancreatic β cells such as glucokinase, *Glut2* (*Slc2a2*), *Kir6.2* but also *Pax4*, a transcription factor involved in the identity of the pancreatic β cell. Thus, demethylases may play a role in inhibiting the expression of genes involved in the identity and function of pancreatic β cells. On the other hand, we demonstrate that siRNAs directed against the methyltransferase *Mettl3* induced a decrease in the expression of *Foxo1* and *MafA* involved in the identity of the pancreatic β cells, and of *Glut2*, involved in the secretion of insulin in response to glucose. Although their precise roles in maintaining β-cell identity remain unknow, recent data suggest that the silencing of methyltransferases impacts the insulin/ IGF1-Akt-Pdx1 pathway [35] with hypomethylation in mRNA of the majority of the genes involved in this pathway, leading to altered β-cell proliferation and a decrease in insulin secretion, which is also a hallmark of T2D. Thus, while m^6^A demethylases may play a role in altering β-cell function, m^6^A methyltransferases may be involved in maintaining the identity and function of pancreatic β cells, probably through the methylation of the mRNAs of these genes. Whether m^6^A methylation of β-cell identity genes controls their fate and/or expression remain to be studied. Interestingly, our data demonstrate that the use the pharmacological FTO inhibitor bisantrene improved β-cell function through an increase of glucose-stimulated insulin secretion. This suggests that FTO may be involved in the downregulation of insulin secretion, and that targeting FTO could improve glucose homeostasis. Recent studies have shown that disabling methyltranserases activity by siRNA or specific β-cell knockout in mice results in reduced m^6^A methylation and impairs insulin secretion [35,36,37]. Based on these observations, we can speculate that the increase in the expression of *FTO* demethylase, as observed in blood cells of T2D patients [39,40] may modulate a set of genes that is also regulated by methyltransferases, finally leading to reduce their methylation and to subsequently impair insulin secretion. Whether m^6^A methylation is directly modulated by bisantrene and/or knock-down of m^6^A enzymes, as well as the genes that are differentially methylated and directly or indirectly affected by the inhibition of FTO activity or expression, remains unknown. Indeed, our data in Min6 cells, mouse and human pancreatic islets treated with bisantrene reveal that the kinetic of FTO inhibition is key to improve GSIS, probably without affecting global m^6^A methylation. A limitation of our study is that the β-cell specific RNAs targeted by m^6^A demethylases and methyltransferases remain to be identified, as well as the contribution of other islet cell types in the glucose-dependent variations of m^6^A methylation. In addition, we observed controversial results using knock-down experiments on β-cell marker expression, such as *MafA* or *Ins1*. We can hypothesize that compensatory mechanisms or m^6^A independent effects may explain these discrepancies. In addition, pharmacological inhibition of FTO or its knock-down, though having similar effect on GSIS, may activate different molecular pathways. Further investigations using targeted qRT-PCR based approaches [54] or untargeted m^6^A sequencing [25,55] are thus required to better understand the molecular mechanisms linking FTO inhibition, m^6^A methylation and β-cell identity gene expression in mouse and human islets. This future data will probably help us to better understand the discrepancies that we identified in this study.

As circulating free fatty acid levels are increased in patients with T2D, we also studied their effects on RNA methylation. We observe that palmitate, in addition to inducing the expression of ER stress genes and to impairing insulin secretion [45], also increases global levels of m^6^A methylation of RNAs. Interestingly, this effect was observed upon chronic high glucose treatment, suggesting that glucolipotoxicity might contribute to modulate m^6^A methylation levels. Based on our results showing a decreased expression of *ALKBH5* demethylase in human pancreatic islets treated with palmitate, this suggests that the increased m^6^A levels upon exposure to palmitate may probably be associated with altered *ALKBH5* expression. The prolonged exposure to a combination of saturated fatty acids and high glucose may contribute to β-cell failure [47]. Intriguingly, when compared to the effects observed in vitro in pancreatic islets treated with glucose and palmitate, we observed decreased global levels of m^6^A methylation of RNAs in pancreatic islets obtained from mouse models of diabetes or T2D donors (Figure 1). These discrepancies might arise from the glucolipotoxic stress by itself, which is different in vitro (i.e., in culture) versus in vivo (i.e., in the whole body). The time of exposure to glucose and FFAs, the nature and concentration of circulating FFAs, the contribution of multi-organ crosstalk, are potential cues that may contribute to the complex nature of this stress that differentially modulates m^6^A methylation of RNAs, as observed in this study. The recent identification of mechanisms and pathways responsible for human β-cell failure upon glucolipotoxicity and recovery of β-cell function provides insights into T2D [56] and may help to elucidate the potential contribution of m^6^A methylation in the regulation of failure and recovery upon glucolipotoxic stress. Although our data concerning the effects of palmitate on m^6^A methylation need to be confirmed in T2D human islets, as well as the effects of other FFAs, further investigations are warranted to better appreciate the pathophysiological contributions of FFAs to m^6^A RNA methylation and their specific target RNAs.

Overall, we have demonstrated that the environment, in particular the glucose concentration, modulates m^6^A methylations and plays a crucial role in the regulation of pancreatic β-cell function, affecting key enzymes involved in the regulation of the epitranscriptome. We have identified that targeting the FTO/m^6^A axis could significantly improve insulin secretion in non-diabetic mouse and human pancreatic islets, suggesting that targeting FTO signaling by effective inhibitors such as bisantrene could be relevant for T2D therapy. Although we did not measure the metabolic effect of bisantrene in preclinical diabetic models such as *Db/Db* mice, our results may contribute to a better understanding of the mechanisms involved in the development of metabolic diseases and identify potential therapeutic targets.

## Figures and Tables

**Figure 1 cells-11-00291-f001:**
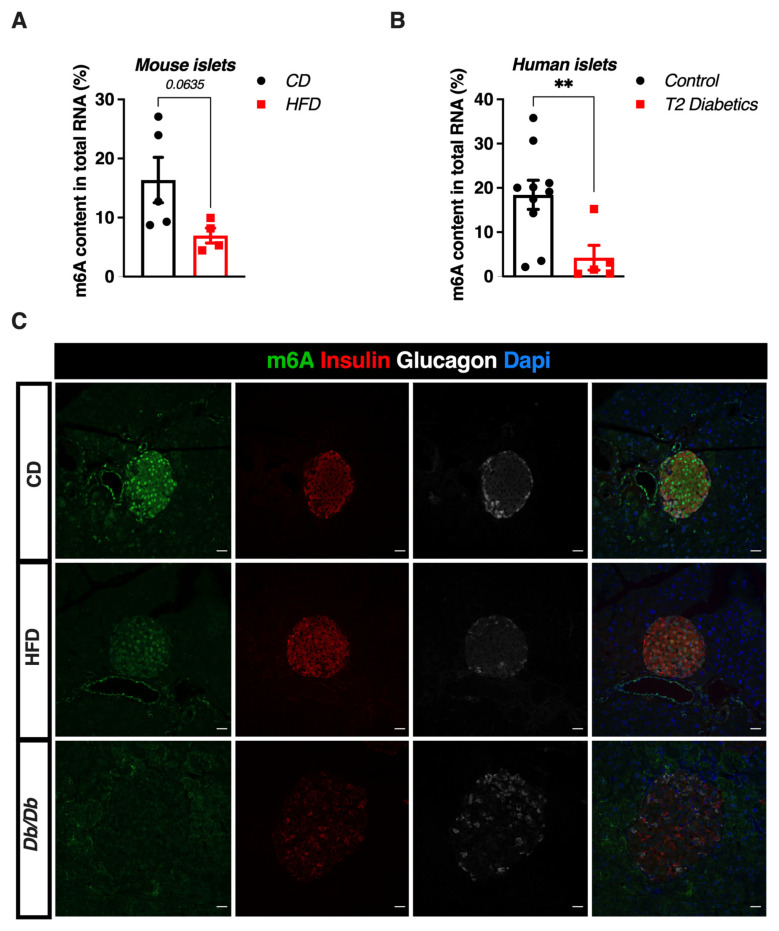
m^6^A RNA methylation in human and murine diabetic islets. (**A**,**B**) ELISA quantification of m^6^A levels in total RNA from (**A**) pancreatic islets isolated from mice fed with chow diet (*n* = 5) or high fat diet during 12 weeks (*n* = 4) and (**B**) human control islets (*n* = 10) vs. human islets from donors with T2D (*n* = 5). Data were analyzed by Mann–Whitney test. ** *p* < 0.01. (**C**) Representative immunofluorescent staining of m^6^A levels (in green), insulin (in red) and glucagon (in white) in pancreatic sections from 15-week-old wild C57Bl6J mice fed with chow (CD), high fat (HFD) diets or from 15-week-old db/db mice. Nuclei were stained with Dapi (in blue). Scale bar represents 22 µm.

**Figure 2 cells-11-00291-f002:**
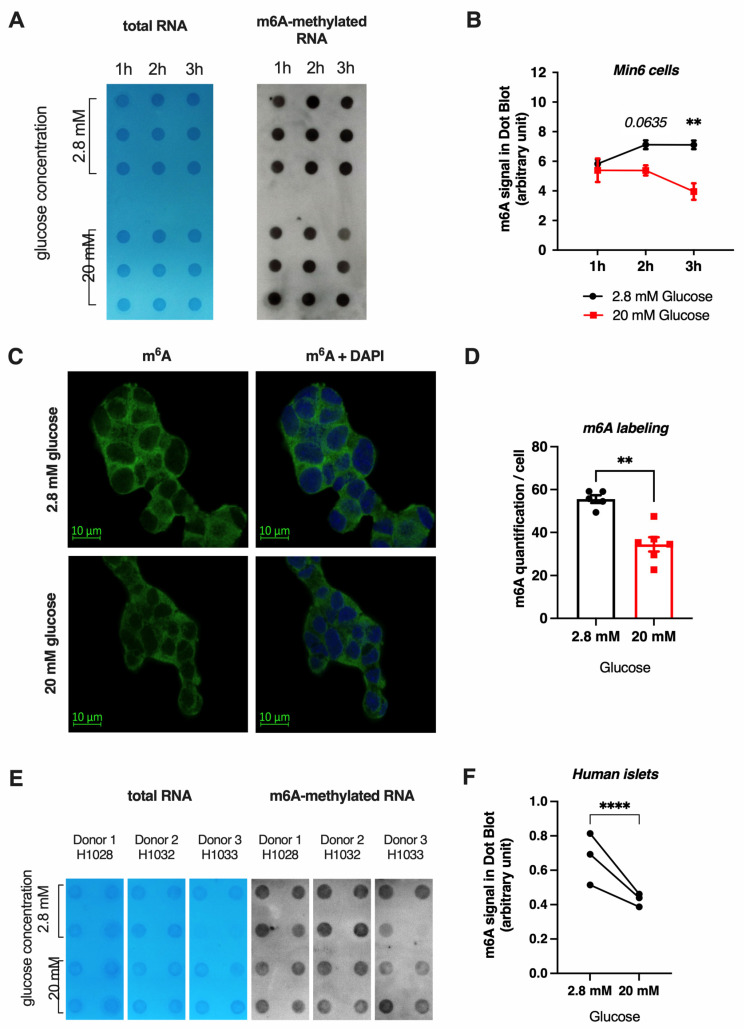
Effect of glucose on m^6^A RNA methylation in Min6 cells and non-diabetic human islets. (**A**,**B**) m^6^A methylation was quantified by dot blot in total RNA of Min6 cells (*n* = 3) after 3 h of treatment with 2.8 or 20 mM glucose. Quantification of m^6^A labeling in B was obtained from nylon membranes ((**A**), right membrane) and normalized by total RNA ((**A**), left membrane). (**C**,**D**) Immunofluorescence of Min6 cells (*n* = 5) after glucose treatment (**C**) and its quantification (**D**). (**E**,**F**) m^6^A methylation was quantified by dot blot in total RNA of non-diabetic human islets from 3 donors (H1028, H1032, H1033) after 1 h of 2.8 or 20 mM glucose treatment (*n* = 3 or 4). Quantification of m^6^A labeling obtained in nylon membrane ((**E**), right membrane) and normalized by total RNA (**E**, left, blue membrane). Data were analyzed by two-way ANOVA with Bonferroni’s correction for multiple comparisons (**B**) or Mann–Whitney tests (**D**,**F**). ** *p* < 0.01, **** *p* < 0.0001.

**Figure 3 cells-11-00291-f003:**
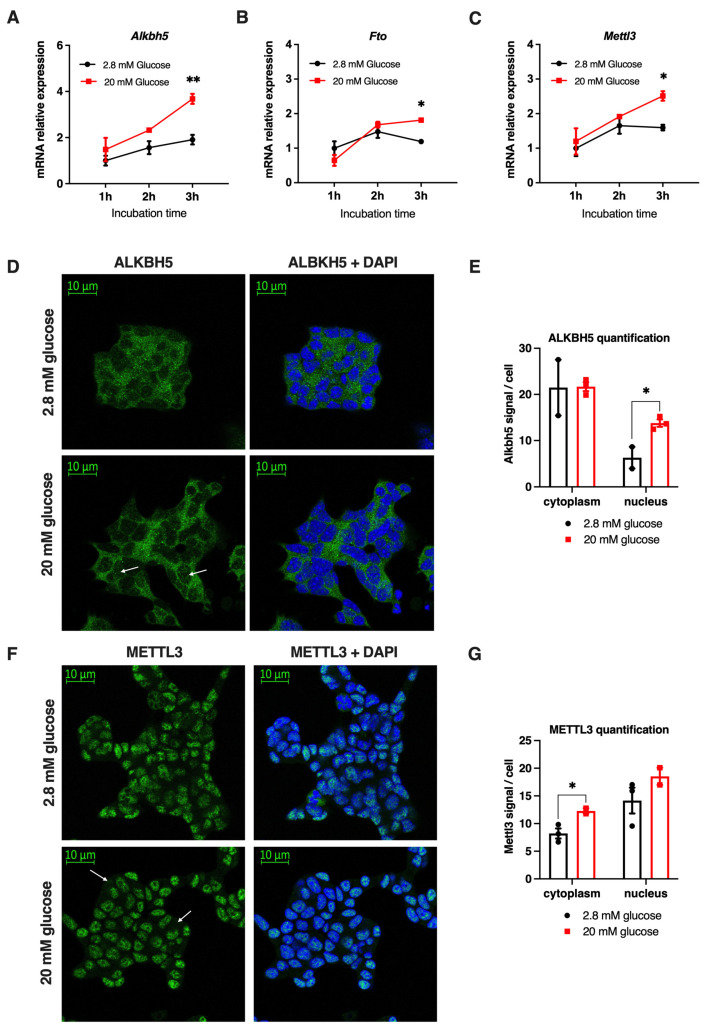
mRNA expression and protein localization of m^6^A reader and erasers after glucose treatment. (**A**–**C**) mRNA expression of *Alkbh5* (**A**), *Fto* (**B**) and *Mettl3* (**C**) were quantified in Min6 cells by RT-qPCR after 1, 2 or 3 h of 2.8 or 20 mM glucose treatment (*n* = 3). Immunofluorescence of ALKBH5 in Min6 cells after 3 h of 2.8 or 20 mM glucose treatment (**D**) and its quantification using ImageJ ((**E**), *n* = 3). Immunofluorescence of METTL3 in Min6 cells after a 3 h treatment with 2.8 or 20 mM glucose treatment (**F**) and its quantification ((**G**), *n* = 3). Data were analyzed by two-way ANOVA with Bonferroni’s correction for multiple comparisons multiple *t*-tests (**A**–**C**) or multiple unpaired *t*-tests (**E**,**G**). * *p* < 0.05, ** *p* < 0.01.

**Figure 4 cells-11-00291-f004:**
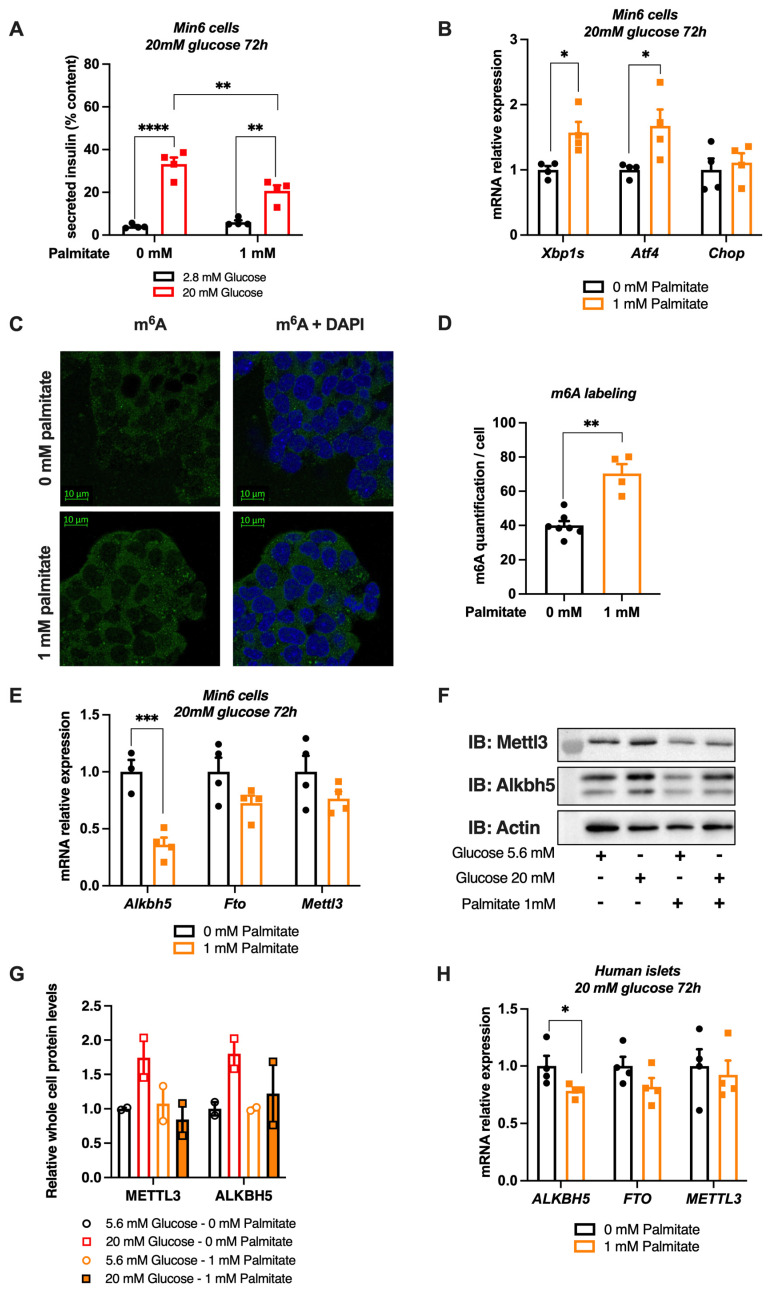
Effects of a chronic high glucose and palmitate treatment on m^6^A RNA methylation and m^6^A enzyme expression in Min6 cells and non-diabetic human islets. (**A**,**B**) Glucose-stimulated insulin secretion was quantified by ELISA ((**A**), *n* = 4) and quantification of mRNA expression levels of some ER stress genes by RT-qPCR ((**B**), *n* = 4) after 20 mM glucose with or without 1 mM palmitate during 72 h. (**C**,**D**) Immunofluorescence of m^6^A methylation in Min6 cells after after 72 h of 20 mM glucose with or without 1 mM palmitate (**C**) and its quantification (*n* ≥ 4, (**D**)). (**E**) Quantification of m^6^A enzyme expression by RT-qPCR in Min6 cells treated with 20 mM glucose with or without 1 mM palmitate ((**E**), *n* = 4). (**F**,**G**) Western blot (**F**) and its quantification (**G**) showing METTL3 and ALKBH5 protein levels in Min6 cells treated with 5.6 or 20 mM glucose, with or without 1 mM palmitate, for 72 h. (**H**) Quantification of m^6^A enzyme expression by RT-qPCR in pancreatic human islets (H1099) treated with 0.5 mM palmitate (*n* = 4). Data were analyzed by two-way ANOVA with Tukey’s correction for multiple comparisons (**A**) or Mann–Whitney tests (**B**,**D**,**E**,**H**). * *p* < 0.05, ** *p* < 0.01, *** *p* < 0.001, **** *p* < 0.0001.

**Figure 5 cells-11-00291-f005:**
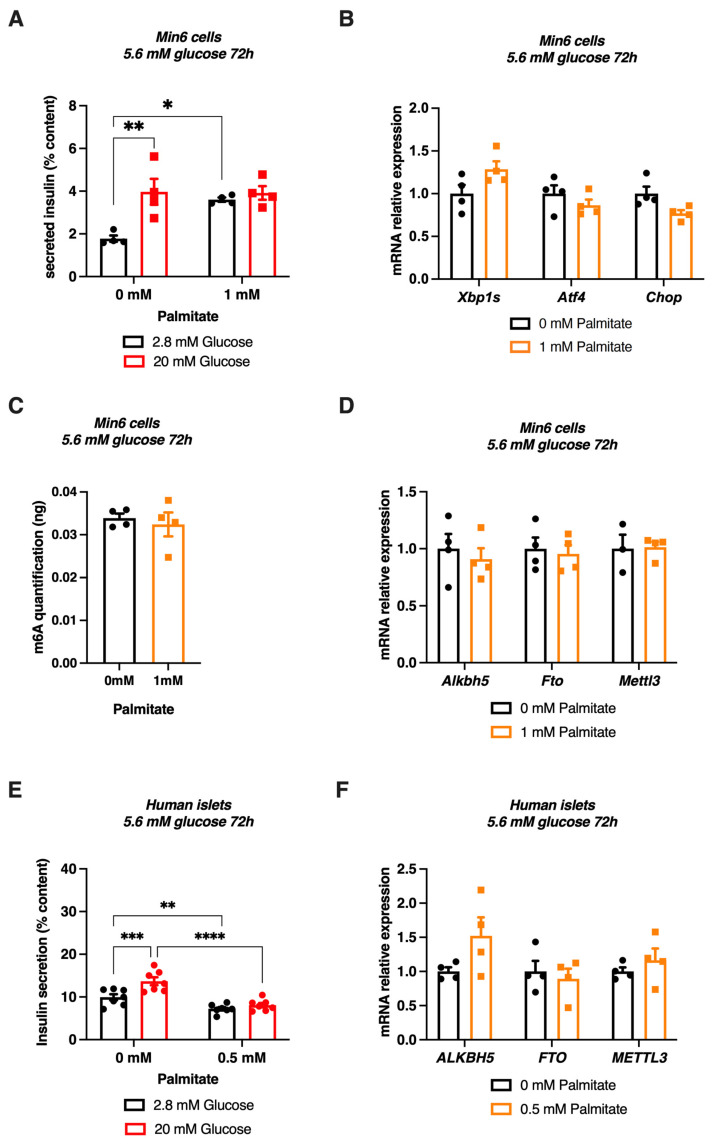
Effects of a chronic low glucose and palmitate treatment on m^6^A RNA methylation and m^6^A enzyme expression in Min6 cells and non-diabetic human islets. (**A**,**B**) Glucose-stimulated insulin secretion was quantified by ELISA ((**A**), *n* = 4) and quantification of mRNA expression levels of some ER stress genes by RT-qPCR ((**B**), *n* = 4) after 5.6 mM glucose and 1 mM palmitate cotreatment during 72 h. (**C**) m^6^A methylation levels in Min6 cells after 72 h of 5.6 mM glucose and 1 mM palmitate ((**C**), *n* = 4). (**D**) Quantification of m^6^A enzyme expression by RT-qPCR in Min6 cells treated with 5.6 mM glucose and 1 mM palmitate (*n* = 4). (**E**) Glucose-stimulated insulin secretion of human islets treated with 5.6 mM glucose and 0.5 mM palmitate was quantified by ELISA. (**F**) Quantification of m^6^A enzyme expression by RT-qPCR in pancreatic human islets (H1099) treated with 0.5 mM palmitate (*n* = 4). Data were analyzed by two-way ANOVA with Tukey’s correction for multiple comparisons (**A**,**E**) or Mann–Whitney tests (**B**–**D**,**F**). * *p* < 0.05, ** *p* < 0.01, *** *p* < 0.001, **** *p* < 0.0001.

**Figure 6 cells-11-00291-f006:**
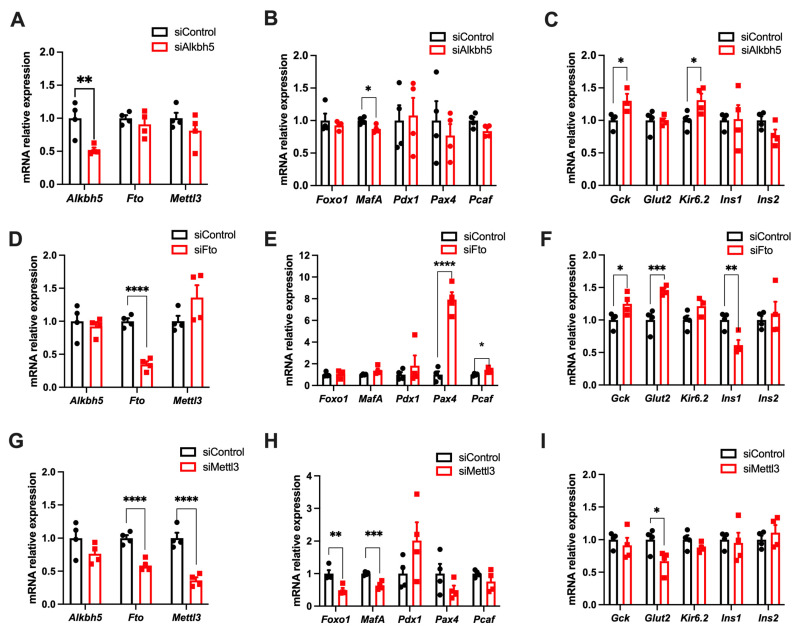
Knock-down of m^6^A enzymes through siRNA affects the expression of genes involved in beta-cell identity and function. Cells were treated during 24 h with a non targetting siRNA (siControl), siAlkbh5, siFto and siMettl3 and lysed 48 h later to study mRNA expression. Quantification of m^6^A enzyme expression by RT-qPCR after siRNA transfection against *Alkbh5* (**A**), *Fto* (**D**) or *Mettl3* (**G**) (*n* = 4). mRNA expression of genes involved in β-cell identity (**B**,**E**,**H**) and function (**C**,**F**,**I**) (*n* = 4) is represented. Data were analyzed by two-way ANOVA with Bonferroni’s correction for multiple comparisons (**A**,**D**,**G**) and multiple t-tests (**B**,**C**,**E**,**F**,**H**,**I**). * *p* < 0.05, ** *p* < 0.01, *** *p* < 0.001, **** *p* < 0.0001.

**Figure 7 cells-11-00291-f007:**
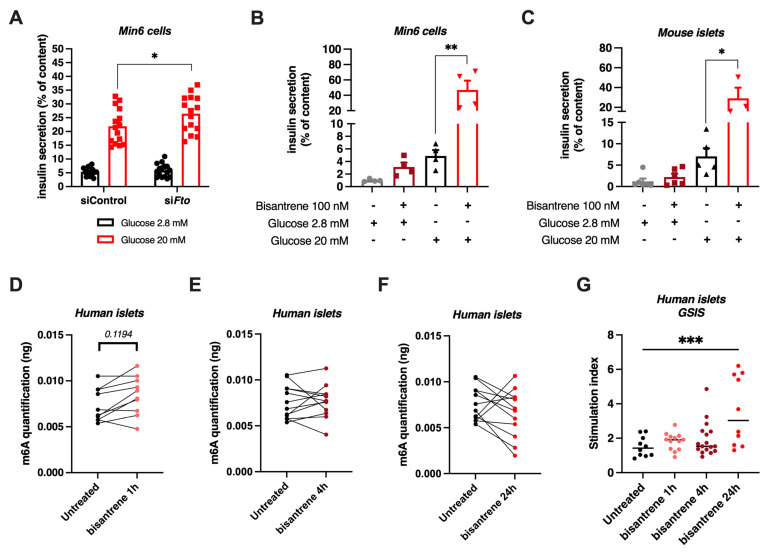
Effects of genetic or pharmacological FTO inhibition on glucose-stimulated insulin secretion in Min6 cells, mouse and human pancreatic islets. (**A**) GSIS after siRNA mediated *Fto* knockdown in Min6 cells (*n* = 16). (**B**,**C**) GSIS after treatment of Min6 cells ((**B**), *n* = 4) and primary mouse pancreatic islets ((**C**), *n* = 6) with 100 nM bisantrene for 8 and 2 h, respectively. (**D**–**F**) Human pancreatic islets were untreated or treated with 100 nM bisantrene for 1 h (**D**), 4 h (**E**) and 24 h (**F**) and global m^6^A RNA methylation was quantified by ELISA. (**G**) GSIS of pancreatic human islets untreated or treated with 100 nM bisantrene for 1, 4 and 24 h. Stimulation index represents the fold of 20 mM glucose-stimulated insulin secretion over 2.8 mM glucose-stimulated insulin secretion. Data were analyzed by two-way ANOVA with Tukey’s correction for multiple comparisons (**A**–**C**), Mann–Whitney tests (**D**–**F**) or one-way ANOVA with Dunnett’s correction for multiple comparisons (**G**). * *p* < 0.05, ** *p* < 0.01, *** *p* < 0.001.

## Data Availability

Not applicable.

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
