# Peer review of "Glucose Regulates m6A Methylation of RNA in Pancreatic Islets"

_cells, 2022, doi:10.3390/cells11020291_

Round 1

Reviewer 1 Report

In this manuscript Bornaque and colleagues evaluate the impact of high glucose and palmitate on m6A RNA methylation and m6A methylase and demethylases expression in pancreatic beta cells. The topic is very interesting and relevant for the diabetes field since m6A RNA changes may affect gene expression and splicing, and such epigenetic alterations have previously been detected in blood from T2D patients.

The manuscript is well written and contains interesting data, however, there are several points that need further clarification or completion before publication, please see below:  

  1. For all the graphics shown in the manuscript please add (within each bar )the data points corresponding to the individual experiments. This is much more informative than showing simple columns since it allows to see the real trend and dispersion of the data.
  2. In Figure 1 A you analyze the m6A in islets from mice fed for 12 weeks with HF diet. Where these mice hyperglycemic compared to the regular chow-fed mice? The metabolic status of these animal must be informed.
  3. According to the Figure legend, Figure 2A shows m6A modification in human islets from 5 type 2 diabetic individuals. However, in Supplementary table 1 only two preparations of diabetic patients are mentioned. This table needs to be completed. In addition, the diabetes duration of the patients is informed for only 1 individual. Moreover, were the control human islets used in these experiments age-matched with the T2D individuals?  Please provide the mean age and SE of each group analyzed.
  4. In Figure 2B the red line corresponding to the m6A DOT BLOT quantification after high glucose treatment does not seem to match the DOT BLOT itself (panel A). The decrease in m6A after 3 h of high glucose treatment is clear only in 2 out of 3 experiments but in the graphic the data looks very consistent, and the error bars are very small. Can you please verify that the quantifications are correct?
  5. Figure 2E shows m6A RNA methylation in 3 human islet preparations exposed or not to 20 mM glucose. These experiments were done in triplicate or quadruplicates. It is not adequate to do statistical analysis on technical replicates. The correct way to represent this experiment is to make only 1 graphic of paired data with 3 points at low glucose and 3 points at high glucose. Each point will be the mean value of the technical replicates done for each of the human islet preparations. To show better the effect of the high glucose treatment you may link the low glucose points with their corresponding high glucose ones. Statistical analysis must be done in n=3 (3 human islet preparations).
  6. The increase in mRNA expression of m6A erasers and writers (Figure3A-C) is also seen at protein level?
  7. Figure 3E shows the quantification of Alkbh5 in the cytoplasm and the nucleus of MIN6 cells after 2.8 and 20 mM glucose. This quantification shows a significant increase of Alkbh5 in the nucleus upon high glucose exposure, however, such increase is undetectable in the picture shown in Figure 3D. The images must be representative of the quantification. Can you please show representative images, increase the zoom of the images and put arrows where you consider that there is an increase in the Alkbh5 nuclear localization? Figure 3F can also be zoomed.
  8. In the methods section and text, it is mentioned that 1 mM or 0.5 mM palmitate coupled with serum albumin was used for palmitate treatment of Min6 cells and human islets, respectively. Which was the albumin and serum in your culture media? This must be clarified because the albumin concentration will determine the concentration of unbound (free fatty acids) in the media, and the FFA present in the serum may also alter the palmitate response.
  9. The effect of palmitate on m6A methylation is measured by immunofluorescence (Figure 4C-D). This method is not very accurate. Dot blots or m6A Elisas must be used for this determination as it was previoulsy done.
  10. Considering that palmitate and high glucose seem to exert opposite effects on m6A RNA methylation, and that in T2D there is a combination both, it will be relevant to assess which is in fact the outcome of a combined treatment high glucose + palmitate on the m6A RNA methylation status of MIN6 cells and human islets.
  11. In Figure 5, several gene expression changes in genes related with pancreatic beta cell function and identity are seen upon Alkbh5, Fto or Mettl3 silencing. How is the m6A RNA methylation status of the cells under these conditions?  This information will be relevant to assess whether the gene expression changes are dependent or independent of alterations in m6A RNA modification. If no changes in total RNA m6A are detected, the authors could look at the m6A methylation status of the modulated gene using m6A RT-qPCR as described by Castellanos-Rubio et al. Sci. Rep. 2019 https://doi.org/10.1038/s41598-019-40018-6 this will be extremely interesting since it will provide information about the influence of m6A methylation on the fate of beta-cell relevant mRNAs.
  12. In Figure 6 it is nicely shown that the Fto inhibitor bisantrene increases insulin secretion in control Min6 cells and non-diabetic mouse islets, however, there is no information on whether this molecule also increases insulin secretion in islets from high fat fed or bd/bd mice. This must be added to justify the conclusion present in lines 438-440 of the manuscript "We have identified that targeting the FTO/m6A axis could significantly improve insulin secretion, highlighting the broad potential of targeting FTO signaling by
    effective inhibitors such as bisantrene for T2D therapy".

Reviewer 2 Report

In this work, Bonarque et al show that glucose and FFA (e.g. palmitic acid) regulate m6A RNA decoration in beta-cells. In particular, they observed a significant decrease of overall m6A levels in T2D islets compared to non-diabetic. In addition, they show that very short (1-3 hr) incubations with 20mM glucose are able to increase the expression levels of methylases (Mettl3) or demethylases (Alkbh5 and Fto) in MIN6 cells. Furthermore, glucose also altered the cellular localization of the enzyme, shuttling methylases in the cytosol and demethylases in the nucleus. Then, the authors measured the expression levels of genes important for beta-cell identity and function in min6 cells ablated of either Alkbh5, Fto or Mettl3. They overall conclusion was that knock downs of demethylases increased the expression of beta-cell identity genes (such as Pax4) and beta-cell function genes (such as Gck, Kir6.2 and Glut2), whereas down regulation of Mettl3 had opposite effect. Finally, they validated these results showing improvement of insulin secretion in Min6 cells and mouse islets treated with Fto inhibitors.

Although the studies are well conducted, I envision some limitations, including the lack of novelty and the scarcity of human data. In addition, the following points require further clarification:

1) The high-fat diet model is commonly considered a model of beta-cell compensation in insulin resistant states. Therefore it cannot be applied to mimic the physiopathology of T2D in humans. The authors should consider assessing m6A levels in beta-cells of db/db mice at late stages of life in which beta-cell fail to compensate, compared to young db/db mice and to age-matched WT animals.

2) Although it has been shown that glucose and other metabolites can regulate gene expression following treatments of a few hours (especially early responder genes), it is unlikely that stimulations of 20 mM glucose cause a ∼2-fold increase of transcript levels of RNA methylases and demethylases. The authors should show also the expression levels of genes which are known to be regulated by glucose at early time points.

3) The changes in gene expression shown following 3 hr stimulation with 20 mM glucose represent acute rather than chronic effects. Therefore such findings can hardly be compared with the events that occur in T2D patients. To evaluate the chronic effects of glucose in m6A regulators, the authors should treat Min6 cells and human islets with 20 mM glucose up to 96 hr and measure the expression levels of Alkbh5, Fto and Mettl3.

4) Although the intracellular redistribution of m6A enzymes in Min6 cells could be appreciated by IF, to have a more reliable quantification the authors should consider repeat these experiments and measure protein levels of the enzymes in the nuclear and cytosolic compartments using cell fractionation methods.

5) The authors reported that FFA have an opposite effect on m6A methylation levels and expression of m6A enzymes compared to the glucose treatments. However, this controversy is not fully discussed. The authors should perform combination treatments to evaluate which mechanism has the major effects on regulating m6A and evaluate gene expression and cellular localization of the methylases and demethylases.

6) Following knockdown experiments, the authors conclude that the ablation of Alkbh5 and Fto increases the expression of genes involved in beta-cell identity and function, whereas Mettl3 KD has opposite effects. However, the authors ignore some controversial data that were not further discussed. Among them, the authors should comment on  why Alkbh5 KD reduces expression levels of MafA. In addition, the authors should discuss why Fto KD decreases Ins1 expression but increases Gck levels. How these results could explain the changes in GSIS shown using Fto inhibitors?

7) The experiments performed using the Fto inhibitors should be repeated in human islets as well.

8) Minor point. In the discussion (lines 418-420) the authors state that "...we can speculate that the increase in the expression of Fto demethylase, as observed in blood sample during T2D [39,40]...", leading the reader to think that Fto is secreted and its circulating levels are altered in T2D. However, the references cited reported that N6 methyladenosine levels were found differentially expressed in the blood of T2D subjects, not Fto. The authors should correct this misleading sentence.

Reviewer 3 Report

The manuscript is interesting and well done. It can be accepted for publication with minor changes. Nevertheless, it would be worth to make some comments:

RESULTS

Data are presented as mean +/- SEM. It seems reasonable to stress the convenience of presenting data as mean +/- SD. See https://journals.physiology.org/doi/full/10.1152/ajpcell.00250.2004

GENERAL MINOR CONCERNS

  • FTO is spelled sometimes as FTO, other times as Fto. The spelling should be unified.
  • The sentence in p.13, l. 365-366 telling that “the diabetic status may contribute to m6A methylation…” is confusing because in the previous line is explaind that m6A methylation is reduced in islets from T2D patients. Then, the diabetic status may contribute to m6A demethylation.
  • It should be explained the reason for choosing palmitate as representative of the FFAA effects. This, because the palmitate action is glucose dependent Carpinelli, A. R., Picinato, M. C., Stevanato, E., Oliveira, H. R., & Curi, R. Insulin secretion induced by palmitate–a process fully dependent on glucose concentration. Diabetes Metab 28(6 Pt 2): 3S37–3S44, 2002). Also, at low-moderate glucose concentrations palmitate is ineffective on electrical activity (Sanchez-Andres JV, Pomares R, Malaisse WJ. Adaptive short-term associativeconditioning in the pancreatic β-cell. Physiol Rep. 2020;8:e14403. https://doi.org/10.14814/ phy2.14403). Such variability cast doubts on the idoneity of palmitate and requires to provide an argument supporting its selection.

Round 2

Reviewer 1 Report

I thank the authors for the effort made to improve the manuscript that is now more comprehensive and complete.

Most of my questions and concerns have been adequately answered, and the related information has been added to the manuscript.

There are however some small things that must be amended:

  1. since the cellular fractionation analysis to evaluate the subcellular localization of METTL3 and ALKBH5 upon high glucose exposure (Supplementary Figure 3) failed to show the same changes seen by immunofluorescence (that is a less exact method to assess that), I suggest softening the statement describing these findings: -Lines 329 to 330: …… “immunofluorescence analysis demonstrated that glucose treatment induced strong variations in the localization of ALKBH5 and METTL3….) please delete STRONG. -Lines 333-334: “Conversely the METTL3 protein, which is mainly nuclear in basal conditions, acquires a cytoplasmic localization upon glucose treatment”, please correct this phrase adding also after the word “acquires”.
  2. This sentence is confusing: Line 321 “did not induce a drop (??) in METTL3 and ALKBH5 protein levels... (did you mean a rise instead of a drop?)
  3. Lines 362-364: “in the absence of palmitate” is written twice
  4. The experiments performed combining high glucose with palmitate showed an increase m6A RNA methylation, which is opposite to what the authors have seen in islets from HFD-fed mice and T2D individuals (two conditions characterized by increased glycemia and saturated FFAs). Even if a paragraph of the discussion is dedicated to the effect of high glucose + palmitate on m6A RNA methylation, the discrepancy between these findings and the reduced m6A levels seen in islets of HFD-fed mice and T2D individuals is not discussed. Please add your interpretation to these opposite findings.

Author Response

Reviewer 1

I thank the authors for the effort made to improve the manuscript that is now more comprehensive and complete.

Most of my questions and concerns have been adequately answered, and the related information has been added to the manuscript.

We would like to warmly thank the reviewer for his/her positive comments concerning the revised version of our manuscript.

There are however some small things that must be amended:

  1. since the cellular fractionation analysis to evaluate the subcellular localization of METTL3 and ALKBH5 upon high glucose exposure (Supplementary Figure 3) failed to show the same changes seen by immunofluorescence (that is a less exact method to assess that), I suggest softening the statement describing these findings: -Lines 329 to 330: …… “immunofluorescence analysis demonstrated that glucose treatment induced strong variations in the localization of ALKBH5 and METTL3….) please delete STRONG. -Lines 333-334: “Conversely the METTL3 protein, which is mainly nuclear in basal conditions, acquires a cytoplasmic localization upon glucose treatment”, please correct this phrase adding also after the word “acquires”.

This has been changed following reviewer’s suggestions.

  1. This sentence is confusing: Line 321 “did not induce a drop (??) in METTL3 and ALKBH5 protein levels... (did you mean a rise instead of a drop?)

The sentence has been corrected.

  1. Lines 362-364: “in the absence of palmitate” is written twice

This has been changed accordingly.

  1. The experiments performed combining high glucose with palmitate showed an increase m6A RNA methylation, which is opposite to what the authors have seen in islets from HFD-fed mice and T2D individuals (two conditions characterized by increased glycemia and saturated FFAs). Even if a paragraph of the discussion is dedicated to the effect of high glucose + palmitate on m6A RNA methylation, the discrepancy between these findings and the reduced m6A levels seen in islets of HFD-fed mice and T2D individuals is not discussed. Please add your interpretation to these opposite findings.

We thank the reviewer for his/her comment. We have now discussed this in the revised version of our manuscript and modified the text as suggested (see paragraph from lines 667 to 675).

Reviewer 2 Report

Thanks to the authors for addressing all the points raised in the previous round, especially for introducing new data in figure 4 and 5.

I just have an additional question on the acute effect of glucose (3 hr) on methylase and demethylase gene expression (former point #2).

Although I appreciate the authors for interrogating early responder genes regulated by glucose, the results showing that none of them is modulated by glucose stimulation raise the question on whether the increased expression of Alkbh5, Fto and Mettl3, shown in Fig. 3A-C,  is either an indirect effect of glucose or perhaps due to other mechanisms that regulate mRNA half-life in the cells. I believe these or other potential explanations should be included in the discussion, and the data on the early responder genes should be included in a supplemental figure.

Author Response

Reviewer 2

Thanks to the authors for addressing all the points raised in the previous round, especially for introducing new data in figure 4 and 5.

We would like to warmly thank the reviewer for his/her positive comments concerning the revised version of our manuscript.

I just have an additional question on the acute effect of glucose (3 hr) on methylase and demethylase gene expression (former point #2).

Although I appreciate the authors for interrogating early responder genes regulated by glucose, the results showing that none of them is modulated by glucose stimulation raise the question on whether the increased expression of Alkbh5, Fto and Mettl3, shown in Fig. 3A-C,  is either an indirect effect of glucose or perhaps due to other mechanisms that regulate mRNA half-life in the cells. I believe these or other potential explanations should be included in the discussion, and the data on the early responder genes should be included in a supplemental figure.

We thank the reviewer for his/her comment. We have now added these data in the supplementary figure 2, and discussed those new results in the revised version of our manuscript (from lines 603 to 609).
